# Machine Learning Applications to Identify Young Offenders Using Data from Cognitive Function Tests

María Claudia Bonfante [1] , Juan Contreras Montes [1,*] , Mariana Pino [2], Ronald Ruiz [2] and Gabriel González [3]

[1] Faculty of Engineering, Institución Universitaria de Barranquilla, Barranquilla 080002, Colombia; maria.bonfante@unisinu.edu.co
[2] Faculty of Psychology, Universidad Autónoma del Caribe, Barranquilla 080020, Colombia; mariana.pino@uac.edu.co (M.P.); ronaldruizp@hotmail.com (R.R.)
[3] Fundación Hogares Claret, Barranquilla 080002, Colombia; gabriel.gonzalez@fhclaret.org
* Correspondence: jcontrerasmontes@gmail.com

**Abstract:** Machine learning techniques can be used to identify whether deficits in cognitive functions contribute to antisocial and aggressive behavior. This paper initially presents the results of tests conducted on delinquent and nondelinquent youths to assess their cognitive functions. The dataset extracted from these assessments, consisting of 37 predictor variables and one target, was used to train three algorithms which aim to predict whether the data correspond to those of a young offender or a nonoffending youth. Prior to this, statistical tests were conducted on the data to identify characteristics which exhibited significant differences in order to select the most relevant features and optimize the prediction results. Additionally, other feature selection methods, such as Boruta, RFE, and filter, were applied, and their effects on the accuracy of each of the three machine learning models used (SVM, RF, and KNN) were compared. In total, 80% of the data were utilized for training, while the remaining 20% were used for validation. The best result was achieved by the K-NN model, trained with 19 features selected by the Boruta method, followed by the SVM model, trained with 24 features selected by the filter method.

**Keywords:** cognitive functions; machine learning; feature selection; violence risk assessment

## 1. Introduction

Young individuals involved in criminal activities during adolescence, known as adolescent offenders, have encountered various legal issues and engaged with the juvenile justice system in their respective countries. This demographic often grapples with a variety of social problems, which include the environment where they live and factors which could generate criminal behaviors [1]. The emotional aspects have been explored, investigating how these individuals process emotions and their subsequent impact on adaptive behaviors and communication with others [2,3].

The manifestation of antisocial behavior in adolescent offenders is linked to frontal lobe damage, with executive dysfunction identified as a risk factor associated with criminal behavior [4]. Language and executive functions are extensively studied cognitive processes concerning adolescent offenders, with indications of challenges in various language aspects, including verbal comprehension and verbal fluency [5,6]. Moreover, difficulties in understanding written information and expressing ideas in writing have been identified [7,8]. Executive function impairments are linked to an increased risk of delinquent behavior in adolescents. It has been found that those with low executive function performance faced a heightened risk of engaging in delinquent and violent behaviors in early adulthood [9,10].

In summary, adolescent offenders confront substantial educational issues, including poor academic performance, dropout, and limited access to suitable educational opportunities [11]. Furthermore, these adolescents could encounter additional societal obstacles,

including racial and ethnic bias, domestic instability, and exposure to community violence. These elements may increase the likelihood that young people will engage in illegal activities and may have an impact on their cognitive function and mental health [12]. These challenges have adverse effects on their academic progress and future prospects. It is imperative to address these issues by developing targeted support and educational interventions to help these individuals to overcome their difficulties and succeed in their educational pursuits.

Consequently, an in-depth exploration of the cognitive processes of adolescent lawbreakers, concerning their educational challenges, becomes crucial. Understanding how these cognitive difficulties impact school performance, decision making, and motivation to learn will pave the way for more effective, tailored educational interventions. Investigating the cognitive processes of these young individuals will form a solid basis for implementing strategies which enhance academic development, improve problem-solving skills, and encourage informed decision making. The study of cognitive processes will play a pivotal role in enhancing educational outcomes and facilitating the successful reintegration of adolescent offenders into society.

In order to obtain information about cognitive processes in young individuals aged between 14 and 18 years, cognitive instruments were administered to two groups of youths: one group comprised young individuals in conflict with the law who were confined in a correctional facility in the city of Barranquilla, Colombia, and another group comprised youths without legal issues. The sample size was constrained by the number of youths confined in the correctional facility. The cognitive instruments employed included the following:

Osterrieth Complex Figure (REY): The REY is a test designed for assessing visuoconstructive ability and visual memory. Additionally, it serves as a tool for evaluating functional impairment across various cognitive dimensions, including attention, concentration, fine motor coordination, visuospatial perception, planning, and spatial orientation.

INECO Frontal Screening (IFS): IFS is a screening test employed to identify executive dysfunction in frontotemporal dementia. It consists of several subtests which target various executive function processes such as motor programming, conflicting instructions, inhibitory motor control, verbal and visual working memory, verbal abstraction ability, and inhibitory control.

Montreal Cognitive Assessment (MOCA): The MOCA is a screening test used to detect mild cognitive impairment. This comprehensive assessment evaluates various cognitive skills, including visuospatial/executive function, naming, episodic memory, attention, language, abstraction, and orientation.

Stroop Color-Word Test: This test is a measure of cognitive conflict and executive control [13]. In addition to assessing processing speed, it also evaluates visual exploration, semantic and phonetic verbal fluency, interference control, and working memory.

Verbal Fluency Test (VFT): This test is designed to assess verbal functioning, and measure verbal ability and levels of executive control.

WAIS: The WAIS is a test that evaluates cognitive abilities and is commonly used to assess IQ. This test comprises 10 subtests focusing on verbal comprehension, perceptual reasoning, working memory, and processing speed. For this study, the matrices, similarities, and vocabulary tasks from the WAIS were utilized.

Symbol Digit Modalities Test (SDMT): The SDMT is employed to identify potential motor or visual difficulties. Additionally, the test assesses attention, short-term memory, and cognitive flexibility.

The main goals of this research can be outlined as follows: (1) Examining the dataset resulting from the above-mentioned cognitive tests to identify the most relevant attributes distinguishing a young individual as either prone or not prone to aggression. (2) Utilizing these crucial traits to evaluate the efficiency of different machine learning algorithms in classifying a study subject as a potential aggressor or nonaggressor. This, in turn, facilitates ongoing research (future endeavors) in the creation of tools for evaluating the risk of violent behavior.

Below is a list of studies related to tools for predicting the risk of violence, especially in the young population.

In several developed countries, such as the United States, youth violence has become the third leading cause of death and the leading cause, among the African American community, in young people aged 10 to 24. For this reason, the Centers for Disease Control and Prevention (CDC) funds Youth Violence Prevention Centers (YVPCs) which use surveillance data to monitor youth violence and assess the impact of their interventions. Access to data is becoming increasingly possible as public health surveillance has mandated the systematic collection and analysis of data [13]. This has generated large volumes of data available to researchers. The use and analysis of data have been crucial in trying to understand the alarming rates of exposure to violence among low-income youth [14].

The assessment of violence risk has become a key element of the criminal justice system. Three benefits can be generated in the application of violence risk assessment tools: (1) results can be used to identify more suitable treatments for the individual; (2) these tools can generate the necessary evidence for the early release of an individual; (3) the tools can determine the timely release of an individual, thereby preventing recidivism [15].

One of the most widely used tools for assessing violence risk is the HCR-20 (Historical Clinical Risk Management-20), which includes 20 risk factors grouped into three categories: Historical (ten factors), Clinical (five factors), and Future Risk (five factors). The HCR-20 generates three risk ratings that can be summarized as low, moderate, or high [16,17].

Another tool that has been used for assessing the risk of violence in hospitalized children and adolescents is the BRACHA method (Brief Rating of Aggression by Children and Adolescents), which has also been used to assess violence risks in schools [18]. This method allows the categorization of children and adolescents into lower or higher risk groups for aggression and violence.

Despite there currently being more than 200 violence risk assessment tools [19], the accuracy of these tools is still below expectations, with a significant presence of false positives and false negatives, with false positives being more common than false negatives [20]. Variables or risk factors associated with a higher probability of an individual acting violently or aggressively include, among others, criminogenic needs, demographic aspects, socioeconomic status, and intelligence. These factors are divided into two categories: static and dynamic. Static factors cannot be changed (age at the time of the first arrest, criminal history, neighborhood, history of abuse, etc.). Dynamic factors can be changed (impulsivity, job skills, drug use, etc.).

Forensic psychiatry is a branch of psychiatry which studies individuals with mental disorders who pose a risk to the public. Forensic psychiatrists support the criminal justice system by investigating the correlation between mental disorders and criminal behaviors. They assess the violence risks of offenders in prisons or secure hospitals and of individuals in the community with mental disorders. Studies conducted in 24 countries, both low- and high-income, showed that one in eight men and one in sixteen women will subsequently commit a serious crime after leaving a psychiatric facility [21]. Due to the high prevalence of criminal acts committed by individuals with severe mental illnesses, the effort to predict the risk of criminal acts by individuals after being discharged from psychiatric facilities is becoming increasingly significant [22].

Some fields of forensic psychiatry, such as the interactions among psychopathology, offense, and aggression, have not been sufficiently investigated, so we are not able to fully understand these interactions. What is clear to the scientific community is that psychiatric illnesses and pathological behavior disorders are not influenced by single and independent factors and cannot be modeled as a linear process. Therefore, and because a large number of psychiatric phenomena do not have monocausal origins but develop from the interactions among various factors, the applications of artificial intelligence, supervised learning, and advanced statistics open up new possibilities in the field of forensic psychiatry. Researchers have studied and compared models to identify the factors that distinguish offender patients with schizophrenia spectrum disorders (SSDs). Using machine learning algorithms applied

to a dataset of 370 patients, researchers achieved the highest accuracy with support vector machines (SVMs), with a balanced accuracy of 77.6% and an AUC of 0.87 [23].

In [24], the enormous potential of the application of machine learning (ML) and deep learning in forensic psychiatry and violence risk assessments is recognized. However, the study also raises the significant risks which can arise from the use of these technologies in decisions to hospitalize or medicate individuals, in sentencing recommendations, or specific police surveillance, because the value systems guiding these technologies are defined by those who design them. As a result, analysis of the ethical implications that may be involved in the use of these techniques is recommended.

Researchers in Denmark developed a supervised ML model designed to assess the risk posed by people in the general psychiatric system according to their future probability of requiring treatment in a forensic institution. The goal was to predict whether people will commit a crime, during or after outpatient treatment or after discharge from hospital care, leading to a court-ordered psychiatric admission in Denmark (through which criminals with mental illnesses are referred for forensic psychiatric treatment). This study used a sample of 45,720 psychiatric patients, of whom 1% committed a crime. The dataset comprised 39 predictors which were divided into three groups: socioeconomic status, psychiatric history, and criminal history. Four ML models were used: logistic regression, random forest (RF), XGBoost, and LightGBM. The performance metrics for each model were presented, including recall, accuracy, and F1-score [25].

In [26], a series of HARM (Hamilton Anatomy of Risk Assessment) models using machine learning techniques was proposed to predict longitudinal physical aggression in patients with schizophrenia in forensic settings. Data from 151 patients were used, with follow-ups at 4, 12, and 18 months. The R language was used, and the following nine machine learning algorithms were implemented: boosted logistic regression, elastic net, lasso regression, k-nearest neighbors, adaptive boosting, extreme gradient boosting, random forest, bagged CART, and conditional forest. The best performance was achieved with the random forest model, with a balanced accuracy of 86.60%, accuracy = 87.33%, and an AUC of 0.914.

Some researchers believe that the results obtained with ML models focus more on predicting the general probability of a crime occurring in a group sample rather than being committed by an individual, and have suggested better statistical strategies. They have developed proposals to predict the type of crime committed by psychiatric patients on an individual level. They conducted tests with models such as random forest (RF), elastic net, and support vector machine (SVM) applied to a representative and diverse sample of 1240 patients from the forensic mental health system. Considering clinical, historical, and sociodemographic variables as predictors, they developed separate models for each type of criminal offense, using feature selection methods. The models showed the following performance: (1) for the prediction of sexual and violent crimes at an individual level, 20 predictor variables were considered, achieving a sensitivity of 83.26% and specificity of 77.42%; (2) for the prediction of sexual and nonviolent crimes at an individual level, 30 predictor variables were considered, achieving a sensitivity of 74.60% and specificity of 80.65%; (3) for the prediction of sexual, violent, and nonviolent crimes at an individual level, 36 predictor variables were considered, achieving a sensitivity of 82.44% and specificity of 60% [27].

A study by [28] focused on defining ML algorithms to obtain models which would identify the factors that distinguish between homicide/involuntary manslaughter and all other crimes committed by individuals with schizophrenia spectrum disorders (SSDs), as well as the factors that distinguish between completed homicide/unpremeditated homicide and other violent crimes. The study concluded that variables related to criminal, psychiatric, and clinical factors have a large influence. They analyzed 358 offender patients with schizophrenia spectrum disorders admitted to the Forensic Inpatient Therapy Center at the Zurich Psychiatry Hospital between 1982 and 2016. Of the total of 358 patients, 36.6% (131) experienced one or more direct coercive measures, while 63.4% (227) did not experience

coercion. The database contained a total of 569 variables. To counteract possible overfitting and improve the model's quality, they reduced the number of variables using the chi-squared test, ultimately obtaining the following 10 most significant variables: (1) threat of violence; (2) (actual) violence towards others; (3) application of direct coercive measures during previous psychiatric hospital treatments; (4) poor impulse control according to the Positive and Negative Syndrome Scale (PANSS); (5) lack of cooperation according to the Positive and Negative Syndrome Scale (PANSS); (6) prescription of haloperidol during hospital treatment; (7) total PANSS score at admission; (8) cumulative daily equivalent antipsychotic dose of olanzapine at discharge; (9) hostility according to the Positive and Negative Syndrome Scale (PANSS); and (10) legal prognosis estimated by a team of licensed forensic psychiatrists. Eight different ML models were trained, with the final naive Bayes model achieving the best results, with a balanced accuracy of 73.28%, an AUC of 0.8468, sensitivity of 72.87%, and specificity close to 73.68%.

The objective of the study conducted by [29] was to identify key differences in offender patients and nonoffender patients with schizophrenia spectrum disorders (SSDs), regarding aggressive behavior, using various supervised ML algorithms. They used a dataset with 740 patients—370 offenders and 370 nonoffenders—and 39 predictor variables: sociodemographic data, psychiatric data, pharmacotherapy, adverse events during hospitalization, childhood or youth abuse data, and other physical or neurological illnesses. The best result was achieved with the "gradient boosting" algorithm, achieving an accuracy of 79.9%, sensitivity of 77.3%, specificity of 82.5%, and an AUC of 0.87.

However, since the brain and behavior are highly complex systems involving multiple levels of temporal and spatial granularity and millions of nonlinear feedback loops, some researchers argue that a better understanding of the common and distinct pathophysiological mechanisms underlying psychiatric disorders is needed to be able to provide more effective and personalized treatments. They also suggest that the analysis of "small" experimental samples using conventional statistical approaches has largely failed to capture the heterogeneity underlying psychiatric phenotypes, so they recommend modern ML algorithms and approaches, such as deep learning, with large volumes of data to achieve better results and a greater understanding of psychiatric phenotypes [30].

Violence specifically within psychiatric hospitalization wards is another globally significant problem that has been studied by the scientific community. In [31], a multivariable prediction model was developed to assess the risk of violence in hospitalized patients, using patient medical record data. The model predicts which patients will exhibit violent behavior during the first 4 weeks after admission. AUC values in the range of 0.797 to 0.764 were achieved.

In [32], a systematic review was conducted, including 182 studies and eight articles, on the use of ML techniques to predict the risk of violence in psychiatric patients in clinical and forensic settings. The researchers analyzed the machine learning methods used in each study, sample size, model performance parameters such as AUC, accuracy, specificity, sensitivity, and the predictors used in each case. The analyzed studies showed AUC values in the range of 0.63 to 0.95.

In (Sonnweber et al., 2022) [33], ML methods were used to identify the factors that distinguish between hospitalized patients with schizophrenia spectrum disorders who consumed alcohol or an illicit substance during forensic psychiatric hospitalization and those who did not. The database consisted of 364 cases, with demographic, clinical, and criminal data from residents at the Zurich Forensic Psychiatry Hospital from 1982 to 2016. Different ML methods were used: logistic regression, decision tree, RF, gradient boosting, k-nearest neighbors, SVM, and naive Bayes. The best performance was achieved with the gradient boosting model, with an AUC of 0.735, sensitivity of 81.48%, and specificity of 57.58%.

This work aims to identify, using ML techniques, the knowledge obtained from databases related to social history, cognitive testing, and risk assessment of young people referred for delinquent behavior. ML techniques are more effective at identifying whether

deficits in cognitive functions contribute to antisocial and aggressive behaviors. This study is motivated by the importance of prevention and of the promotion of mental health services and tools that enable the effective treatment of individuals so that they can reintegrate into community life, reduce the risk of recidivism, and reduce the substantial costs associated with incarceration.

## 2. Materials and Methods

### 2.1. Dataset

Our study sample included adolescent offenders ($n$ = 66) and adolescent nonoffenders ($n$ = 62) of the male gender aged 14–18 years. The offender group was selected from "Centro de Reeducación el Oasis" through the foundation "Hogares Claret" in Barranquilla, Colombia, where the subjects were imprisoned for violation of punishable offenses such as sexual abuse, homicide, and theft, among others. The control group was selected from various educational institutions located in the same city and had to meet the following criteria: (a) male gender; (b) aged between 14 and 18 years old; (c) not more than 12 years of education; (d) absence of criminal background; (e) absence of neurological, psychiatric, or physical diagnosis. The parents or legal representatives of subjects in both groups (adolescent offenders and adolescent nonoffenders) were duly informed about the research by means of an informed consent form which was filled out voluntarily or, in the case of participants under 18 years of age, by parents or guardians. This study was approved by the Caribbean Ethics Committee and followed the ethical principles of the Declaration of Helsinki.

The dataset was built using the following cognitive instruments: Osterrieth Complex Figure; INECO Frontal Screening; Montreal Cognitive Assessment; Stroop Color-Word Test; WAIS; and Symbol Digit Modalities Test. The final dataset contains 138 observations and 39 features, as shown in Table 1.

**Table 1.** Features.

| Cognitive Tests | Features |
|---|---|
| Osterrieth Complex Figure | Copying |
| | Short-term memory |
| | Long-term memory |
| INECO Frontal Screening (IFS) | Motor programming |
| | Interference resistance |
| | Motor inhibitory control |
| | Verbal inhibitory control |
| | Verbal working memory |
| | Numerical working memory |
| | Visual working memory |
| | Abstraction capacity |
| | IFS total |
| Montreal Cognitive Assessment (MOCA) | Executive visuospatial |
| | Identification |
| | Memory |
| | Attention |
| | Language |
| | Abstraction |
| | Orientation |
| | MOCA Total |

**Table 1.** *Cont.*

| Cognitive Tests | Features |
| --- | --- |
| STROOP | Word |
| | Color |
| | Word-color |
| VFT (Verbal Fluency Test) | Phonological fluency success |
| | Phonological fluency repetitions |
| | Phonological fluency distortions |
| | Semantic fluency success |
| | Semantic fluency repetitions |
| | Semantic fluency distortions |
| | Exclusive fluency success |
| | Exclusive fluency repetitions |
| | Exclusive fluency distortions |
| WAIS (Wechsler Adult Intelligence Scale) | Matrices |
| | Similarities |
| | Vocabulary |
| SDMT (Symbol Digit Modalities Test) | Total correct answers |
| | Total answers |

The dependent variable is named "Group" and has two classes: adolescent offenders (SG) and adolescent nonoffenders (CG), as shown in Figure 1.

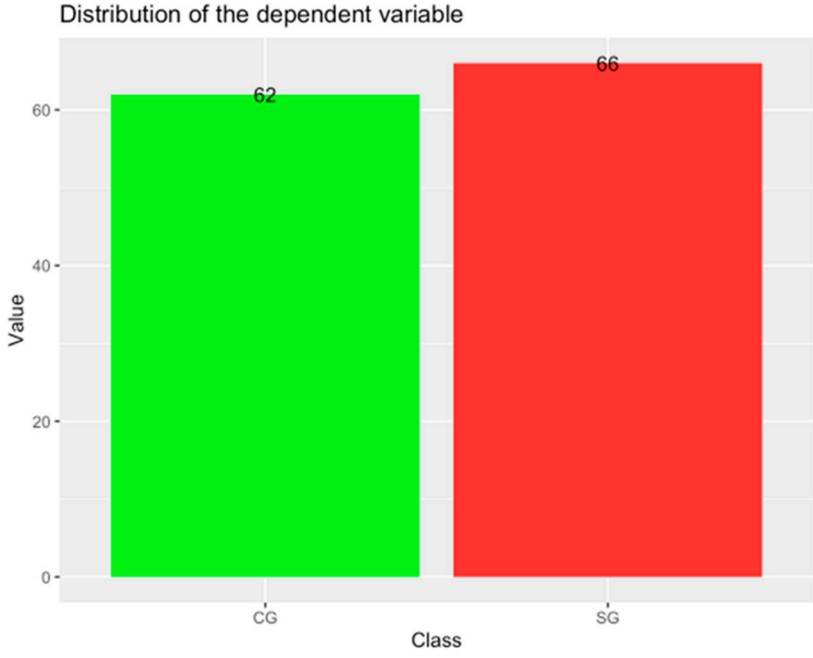

**Figure 1.** Distribution of classes (groups).

The class SG contains 66 observations, and the class CG contains 62 observations, as shown in Figure 1.

## 2.2. Feature Selection

In order to reduce the number of input variables, select the most relevant features, and prevent and reduce overfitting in the machine learning training process, four methods of feature selection were considered. Feature selection methods were applied after splitting the dataset into training and test dataset (80/20), because the test set aims to be a brand-new set on which the machine learning models will be evaluated.

### 2.2.1. Features That Are Significantly Different among Groups

By identifying the most important features that are significantly different among each group, it is possible to develop a more accurate model which can accurately predict the class labels for new data. To select features, the procedure consisted of identifying those characteristics that presented significant differences among each of the groups (classes). The homogeneity of variance test method used was the Levene test. As a result, 23 features with $p < 0.05$ were extracted from the original 36 features. These 22 features are shown in Figure 2.

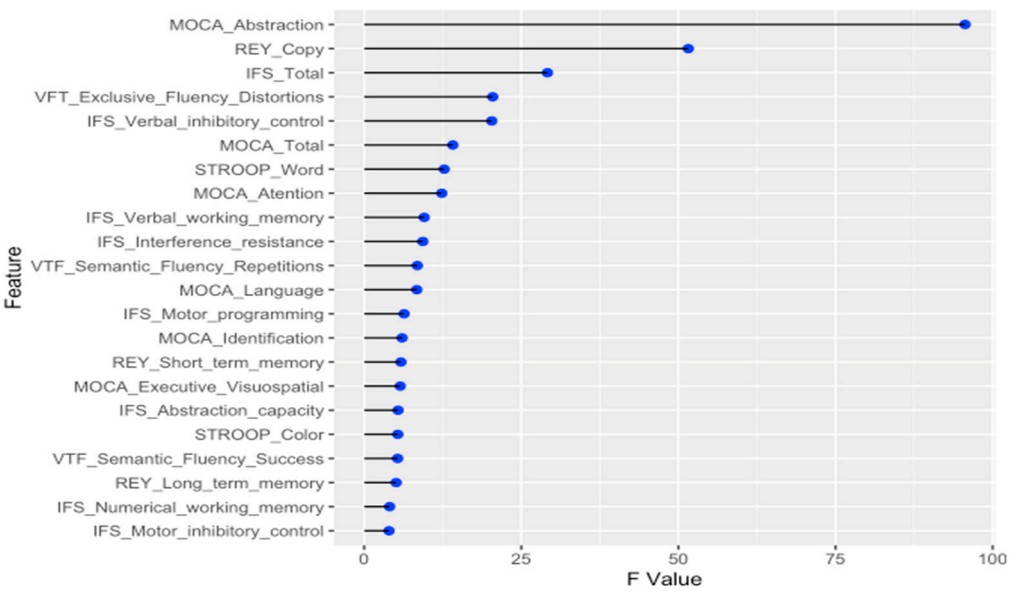

**Figure 2.** Features that are significantly different among groups.

### 2.2.2. Boruta Algorithm

Boruta is a feature selection algorithm that works as a wrapper algorithm around random forest. It operates by extracting features from the initial dataset and generating a duplicate set of these features. In this duplicated set, the values in each column undergo shuffling to introduce randomness, resulting in what are termed shadow features. Subsequently, these shadow features are combined with the original features, forming a new feature space with dimensions twice those of the original dataset. The algorithm then constructs a classifier, specifically a random forest classifier, on this expanded feature space. The classifier assesses the importance of features using a statistical test. Here is where the evaluation takes place: The algorithm compares the importance of the real (original) feature with the maximum importance observed among the shadow features. If the real feature surpasses the importance of the shadow features, it is deemed significant and retained; otherwise, if it is considered insignificant, it is excluded from the dataset. The features deemed significant in the first iteration constitute the dataset used in the subsequent iteration. The algorithm repeats this process by once again creating shadow features based on the features confirmed in the previous iteration. Their importance is evaluated similarly to the first iteration. This iterative cycle continues until a specified number of iterations is reached, or when all features have either been confirmed or dropped. The strictness of the algorithm can be adjusted by modifying the *p*-values and the number of times the algorithm is run. The 19 features selected are shown in Figure 3.

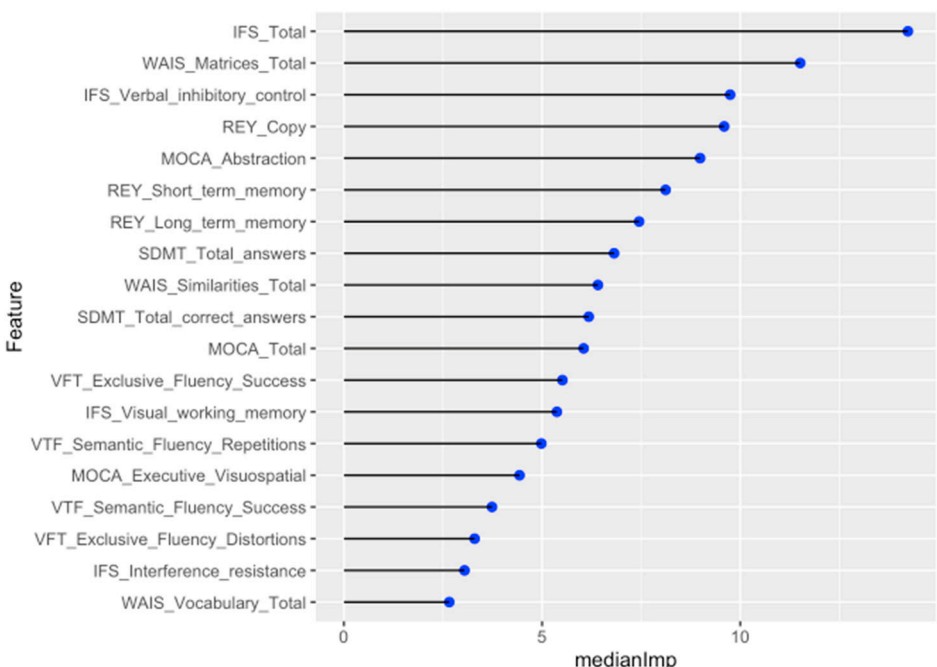

**Figure 3.** Features selected by Boruta algorithm.

### 2.2.3. Recursive Feature Elimination (RFE) Algorithm

The recursive feature elimination (RFE) algorithm works as a wrapper method, systematically discarding features and constructing a model with the retained features. The process involves ranking features according to their importance and iteratively removing the least significant ones until the desired number of features is achieved. The 14 features selected are shown in Table 2.

**Table 2.** Features selected by the RFE algorithm.

| Cognitive Tests | Features | Ranking |
|---|---|---|
| Osterrieth Complex Figure | Copying | 4 |
| | Short-term memory | 7 |
| | Long-term memory | 9 |
| INECO Frontal Screening (IFS) | Verbal inhibitory control | 3 |
| | Visual working memory | 13 |
| | IFS total | 1 |
| Montreal Cognitive Assessment (MOCA) | Abstraction | 5 |
| | MOCA total | 12 |
| VFT (Verbal Fluency Test) | Semantic fluency repetitions | 11 |
| | Exclusive fluency success | 14 |
| WAIS (Wechsler Adult Intelligence Scale) | Matrices_Total | 2 |
| | Similarities_Total | 10 |
| SDMT (Symbol Digit Modalities Test) | Total correct answers | 8 |
| | Total answers | 6 |

### 2.2.4. Filter Algorithm

Filter methods select features based on their scores in various statistical tests to determine their correlation with the outcome variable. One of the main advantages of the filter method is that it is independent of any machine learning algorithm. In this case, we used the filterVarImp function in R (from the Caret library), which uses ROC curve analysis on each predictor and area under the curve as scores. This function can be used to evaluate

features without the need for a specific model. The selection of features is independent of any machine learning algorithms. The score for each feature is shown in Figure 4.

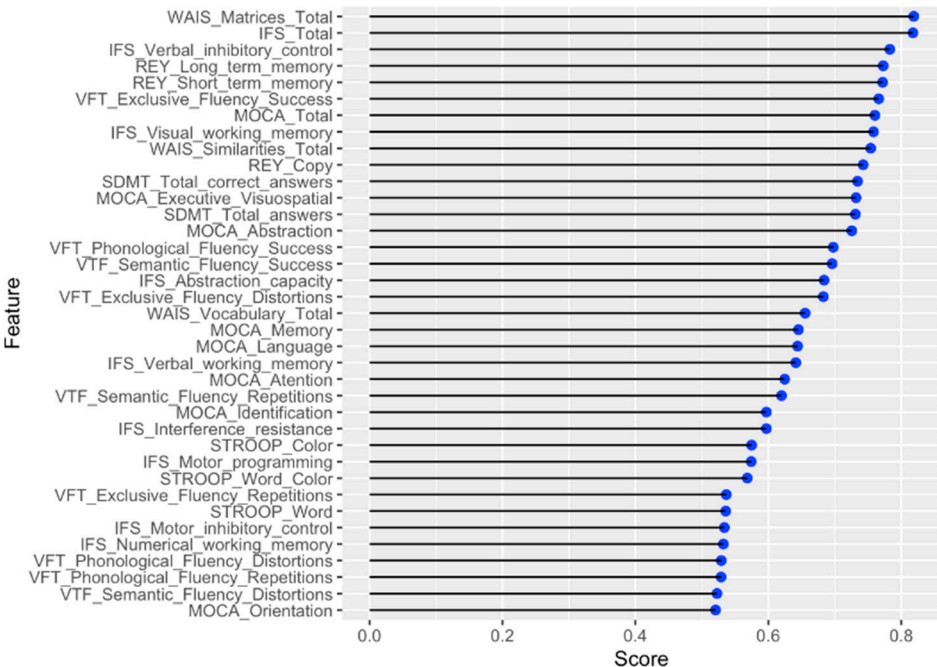

**Figure 4.** Score of features according to filterVarImp function in R.

In this case, the features with a score higher than 0.6 were selected.

## 3. Results

The next step was training and testing three machine learning models using the features selected by each feature selection method. In this study, the dataset was divided into 80/20 (train/test), and cross-validation was used to estimate prediction error in the training subset. Results are shown in Table 3.

**Table 3.** Metric results by machine learning models.

| Model | Metric | All Features | Levene Test | Features Selected by Boruta | RFE | Filter |
|---|---|---|---|---|---|---|
| SVM | Sensitivity/recall | 1.0000 | 1.0000 | 1.0000 | 0.9167 | 0.9167 |
| | Specificity | 0.6923 | 0.6154 | 0.5385 | 0.7692 | 0.8462 |
| | F1 | 0.8571 | 0.8276 | 0.8000 | 0.8452 | 0.8800 |
| | Balance accuracy | 0.8462 | 0.8077 | 0.7692 | 0.8429 | 0.8814 |
| RF | Sensitivity/recall | 0.8333 | 0.6667 | 1.0000 | 0.7500 | 0.8333 |
| | Specificity | 0.8462 | 0.7692 | 0.6923 | 0.6923 | 0.7692 |
| | F1 | 0.8333 | 0.6957 | 0.8571 | 0.7200 | 0.8000 |
| | Balance accuracy | 0.8397 | 0.7179 | 0.8462 | 0.7212 | 0.8013 |
| KNN | Sensitivity/recall | 0.9167 | 1.0000 | 1.0000 | 0.9167 | 0.9167 |
| | Specificity | 0.6923 | 0.4615 | 0.7692 | 0.8462 | 0.6923 |
| | F1 | 0.8148 | 0.7742 | 0.8889 | 0.8800 | 0.8148 |
| | Balance accuracy | 0.8045 | 0.7308 | 0.8846 | 0.8814 | 0.8045 |

The best results were achieved with the K-NN model trained with features selected by the Boruta method (balance accuracy = 0.8846) followed by the K-NN model trained with features selected by the RFE method (balance accuracy = 0.8814) and the SVM model trained with features selected by the filter method (balance accuracy = 0.8814). Other comparison metrics are shown in Table 3 (sensitivity/recall, specificity, and F1-score).

## 4. Discussion

The main objective of this study was to identify young law offenders from nonoffenders based on a dataset extracted from cognitive tests such as the Osterrieth Complex Figure (REY), INECO Frontal Screening (IFS), Montreal Cognitive Assessment (MOCA), Stroop Color-Word Test, WAIS, and Symbol Digit Modalities Test (SDMT). These cognitive tests were administered to a group of young individuals consisting of 66 law offenders and 62 nonoffenders.

In total, the dataset contained 39 features and 138 observations. Different methods were applied to reduce the number of features. Initially, a total of 22 features that showed significant differences between both groups were extracted. Subsequently, feature selection methods such as Boruta (19 features), RFE (14 features), and filter (24 features) were applied.

The Figure 5 shows the number of times each feature was selected by each of the methods employed:

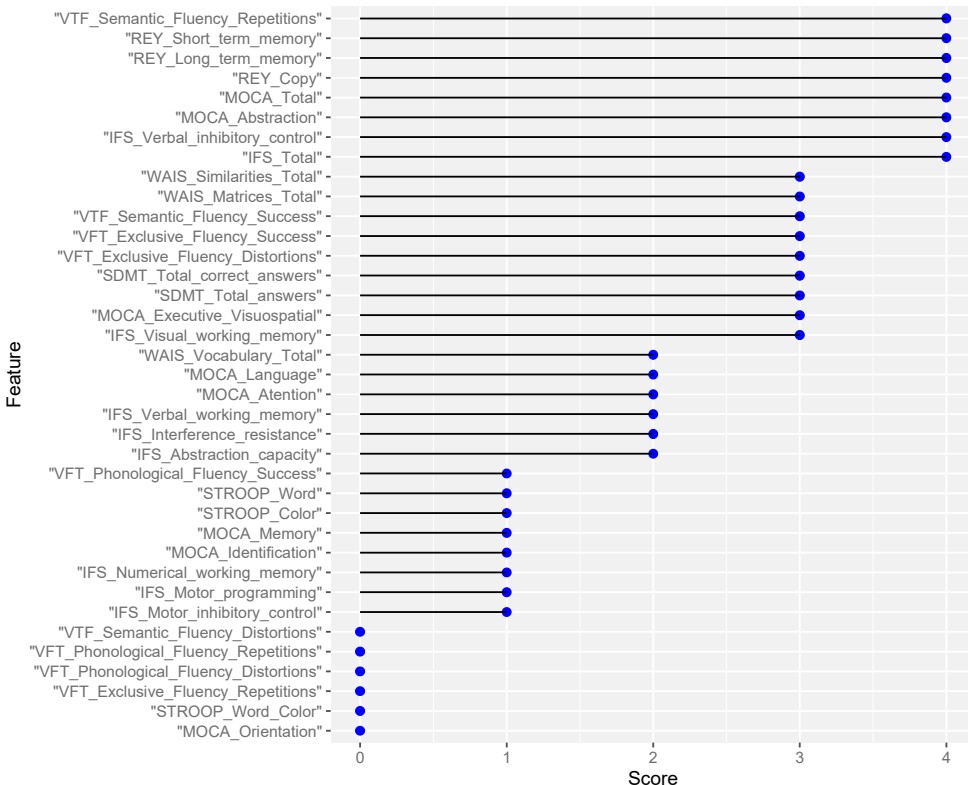

**Figure 5.** Number of times each feature was selected by all methods.

In the Table 4, the eight common features that were selected by the four aforementioned methods are presented.

**Table 4.** Eight common features selected by each of the four methods.

| Cognitive Tests | Features |
| :---: | :---: |
| Osterrieth Complex Figure | Copying<br>Short-term memory<br>Long-term memory |
| INECO Frontal Screening (IFS) | Verbal inhibitory control<br>IFS total |
| Montreal Cognitive Assessment (MOCA) | Abstraction<br>MOCA total |
| VFT (Verbal Fluency Test) | Semantic fluency repetitions |

In the Table 5, the 14 common features that were selected by the three feature selection methods (Boruta, RFE, and filter) are presented.

**Table 5.** Fourteen common features selected by each of the three feature selection methods.

| Cognitive Tests | Features |
|---|---|
| Osterrieth Complex Figure | Copying<br>Short-term memory<br>Long-term memory |
| INECO Frontal Screening (IFS) | Verbal inhibitory control<br>Visual working memory<br>IFS total |
| Montreal Cognitive Assessment (MOCA) | Abstraction<br>MOCA total |
| VFT (Verbal Fluency Test) | Semantic fluency repetitions<br>Exclusive fluency success |
| WAIS (Wechsler Adult Intelligence Scale) | Matrices<br>Similarities |
| SDMT (Symbol Digit Modalities Test) | Total correct answers<br>Total answers |

According to the feature selection methods employed, the three most relevant features are as follows: IFS_Total, WAIS_Matrices_Total, and IFS_Verbal_Inhibitory_control.

The summary of the results obtained for each model is presented in Figures 6 and 7.

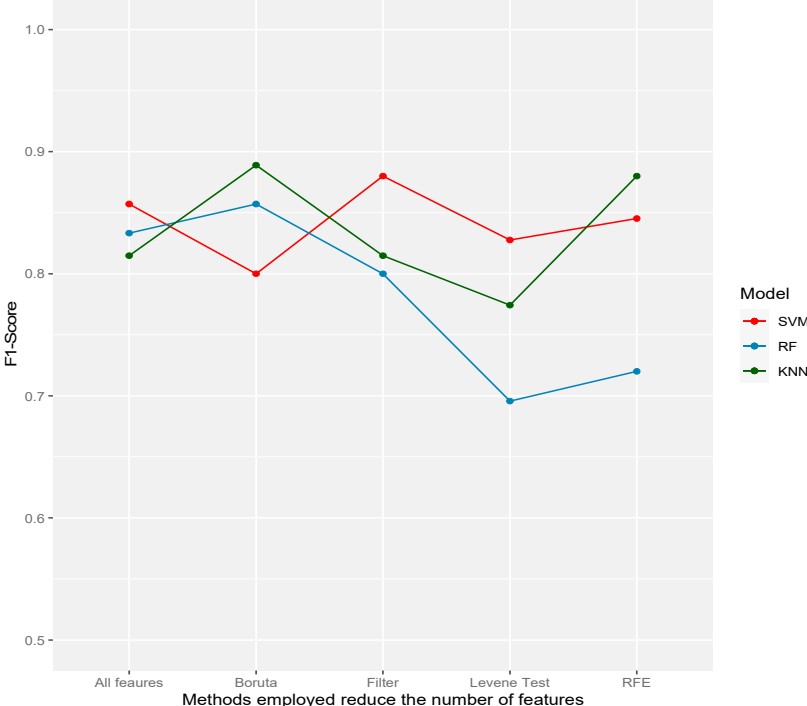

**Figure 6.** F1-score obtained for each model on the testing data.

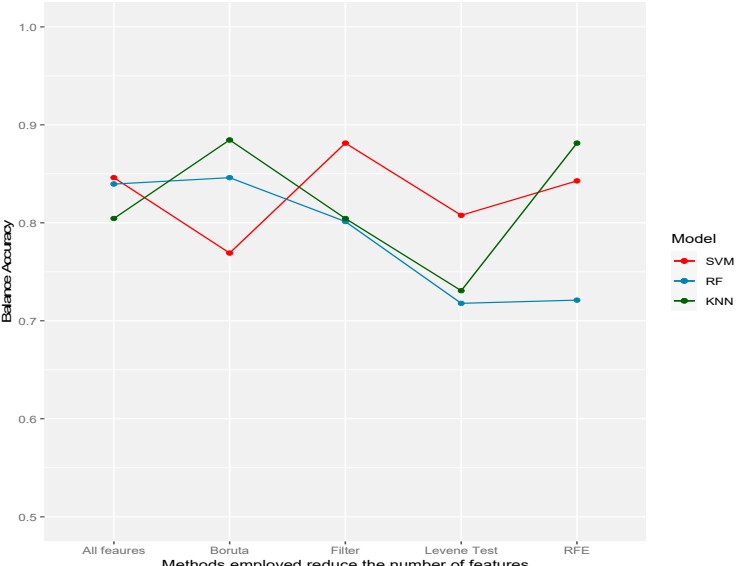

**Figure 7.** Balance accuracy obtained for each model on the testing data.

## 5. Conclusions

In this study, a database containing the results of cognitive tests conducted on a study group (SG) consisting of 66 young offenders in a reform school in the city of Barranquilla, northern Colombia, and a control group (CG) consisting of 62 nonoffender high school students was used. All the young individuals were male and aged between 14 and 18 years old. The database has a total of 37 predictor variables.

The objective was to find a model to predict which group each young individual belongs to based on the variables resulting from the cognitive tests. Three models were employed: support vector machine (SVM), random forest (RF), and k-nearest neighbors (K-NN). Each of these models was trained using a different number of variables, as follows: (1) all 37 variables; (2) only the 22 variables that showed significant differences (Levene Test); (3) only the 19 variables selected by the Boruta method; (4) only the 14 variables selected by the RFE method; (5) only the 24 variables selected by the filter method.

The best result was achieved by the K-NN model trained with the 19 features selected by the Boruta method, followed by the K-NN model trained with the 14 features selected by the RFE method, and, thirdly, by the SVM model trained with the 24 features selected by the filter method.

The three most relevant features are as follows: IFS_Total, WAIS_Matrices_Total, and IFS_Verbal_Inhibitory_control.

As mentioned, the sample size was constrained by the number of youths confined in the correctional facility. As a result, this study is based on a relatively small sample size, which limits generalizability. Therefore, the authors recommend validation of these findings in further studies, preferably with larger populations, and, if feasible, exploring new technologies such as parallel machine learning [34].

**Author Contributions:** Conceptualization, M.C.B., J.C.M. and M.P.; methodology, J.C.M. and M.C.B.; software, J.C.M.; validation, M.P. and R.R.; formal analysis, R.R.; investigation, M.C.B., J.C.M. and M.P.; resources, M.P. and R.R.; data curation, J.C.M.; writing—original draft preparation, M.C.B. and J.C.M.; writing—review and editing, M.P. and R.R.; visualization, M.C.B. and J.C.M.; supervision, M.P and G.G.; project administration, M.P. and G.G.; funding acquisition, G.G. All authors have read and agreed to the published version of the manuscript.

**Funding:** This research and APC was funded by Institución Universitaria de Barranquilla and Ministerio de Ciencia Tecnología e Innovación in Colombia, grant number 68229 BPIN 20200000100006.

**Informed Consent Statement:** Informed consent was obtained from all subjects involved in the study.

**Data Availability Statement:** The data presented in this study are openly available in Kaggle at https://www.kaggle.com/datasets/mariaclaudiabonfante/funcionescognitivas (accessed 20 September 2023).

**Acknowledgments:** We would like to extend our gratitude for the support received from the Ministerio de Ciencia Tecnología e Innovación—MinCiencias, Colombia; the Fundación Clarets; the Centro de Reeducación el Oasis; and the Fundación Luz Esperanza in Barranquilla, Colombia. Their invaluable contributions significantly facilitated the realization of this research.

**Conflicts of Interest:** The authors declare no conflict of interest.

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
