# Peer review of "Machine Learning Applications to Identify Young Offenders Using Data from Cognitive Function Tests"

_data, 1980_

Round 1

Reviewer 1 Report

Comments and Suggestions for Authors

This study presents the results of cognitive tests conducted on young offenders and non-offenders in Colombia using different machine learning models and finds that the KNN model trained with 19 features selected by the Boruta method and the SVM model trained with 21 features selected by the filter method show the best performance. Generally, the objective of the study is clear and the results have some reference value for other researches. Followed are some suggestions for the authors to revise the paper.

1. The innovation of the study should be proposed more clearly. 

2. I think some more discussions on the results are also necessary. For example, the authors only tell us that the KNN method with 19 features is the best, but in my opinion, the 19 features are also deserved to be discussed to some extent, since it has some consultant value for understanding the impact of cognitive function on crime.

3. Figure 1 should be change to a table with more information.

Reviewer 2 Report

Comments and Suggestions for Authors

This paper employs Machine Learning techniques to assess whether cognitive deficits play a role in antisocial and aggressive behavior. It presents the results of cognitive function tests conducted on delinquent and non-delinquent youths, using a dataset with 37 predictor variables and one target variable. Three algorithms are trained to predict youth offender status, with feature selection methods like Boruta, RFE, and Filter applied. The K-NN model, trained with 19 Boruta-selected features, yields the best result, followed by the SVM model with 24 Filter-selected features.

I commend the authors for this work. However, multiple issues were observed.

1. Please remove "." (beside target) in the fourth line of the abstract. It is cutting the sentence flow.

2. I also observed that the introduction is mixed with related work. A general "introduction" should focus on framing the research area, giving a high-level view of the subject matter without diving into specific details about previous research. I recommend dividing your introduction section into related work so that it helps users understand broader research and why your work is important and the related work delves deeper into what specific studies are performed in this area and specific gaps that need to be addressed. You can connect if your work is building on this previous literature or filling the research gaps.

3. Please improve the quality of Figure 2. It is blurry.

4. Please provide details on when you split the dataset. Did you do it just before training and testing ML models? If that is the case, the feature selection should be applied only on training data and not whole data. This will bias your model performance.

5. It also looks like this small dataset is divided into 80/20 (train/test). How can we be sure that your results are generalizable? Why not try cross-validation?

6. Its also unclear how the models trained on these lower samples can provide insights that are generalizable and apply to population-level data.

Overall, the work has fundamental issues and also limited novelty.

Comments on the Quality of English Language

NA

Reviewer 3 Report

Comments and Suggestions for Authors

-The introduction provides a clear context for the study, highlighting the importance of identifying cognitive function deficits in relation to antisocial behavior. It effectively outlines the research problem and objectives. However, it could benefit from a concise literature review on the intersection of cognitive function and delinquency.

-The methodology section describes the approach used, including cognitive function assessments, feature selection methods (Boruta, RFE, Filter), and machine learning algorithms (SVM, RF, KNN). While the methodology is clear, it lacks details on the specific cognitive function tests used and their relevance to delinquency prediction. Further explanation of how the Boruta, RFE, and Filter methods work would enhance clarity.

-The paper reports that the K-NN model trained with features selected by the Boruta method achieved the best results. However, the results lack context and detailed statistical analysis. Including metrics such as accuracy, precision, recall, and F1-score would provide a more comprehensive evaluation of model performance.

-The discussion provides insights into the effectiveness of different feature selection methods and machine learning algorithms. However, it lacks a deeper analysis of the implications of these findings for identifying and addressing delinquent behavior in youths. A discussion of potential biases in the data and model limitations would strengthen this section.

-The paper should be enriched with references to existing research papers on parallel machine learning such as https://doi.org/10.58496/MJBD/2023/002. This would strengthen the argument and provide readers with additional resources for further exploration.

-To enhance the paper's impact, the author should discuss potential future directions for this research.

Round 2

Reviewer 2 Report

Comments and Suggestions for Authors

Please proofread before final submission. The article has repeated statements like the one below.

"Boruta is a feature selection algorithm that works as a wrapper algorithm around Random Forest. Boruta algorithm works as a wrapper algorithm around Random Forest."

There are a lot of grammatical mistakes as well that need to be corrected. See the below paragraph in the conclusion that the authors added

"How it was mentioned, the sample size was constrained by the number of youths confined in the correctional facility. Because that, this study is based on a relatively small sample size, which limit generalizability. Therefore, the authors recommend validation of these findings in further, preferably larger populations and, If feasible, exploring new technologies as parallel machine learning [34]."

  •  
Comments on the Quality of English Language

There is a need to improve English in this article. There are a lot of grammatical mistakes which make sentences hard to comprehend. I recommend authors utilize tools like Grammarly or any other that helps them correct issues in the article.

Author Response

We really appreciate for your precious time in reviewing our paper and providing valuable comments. The manuscript was revised by MDPI English Editing, following the suggestions, and the grammar errors were corrected appropriately. A new version has been submitted. Thanks a lot. 

Reviewer 3 Report

Comments and Suggestions for Authors

Accepted

Author Response

(The authors gave the same response as above.)
